# AdaCL: Adaptive Continual Learning

**Elif Ceren Gok Yildirim**    **Murat Onur Yildirim**    **Mert Kilickaya**    **Joaquin Vanschoren**

Automated Machine Learning Group, Eindhoven University of Technology

## Abstract

Class-Incremental Learning aims to update a deep classifier to learn new categories while maintaining or improving its accuracy on previously observed classes. Common methods to prevent forgetting previously learned classes include regularizing the neural network updates and storing exemplars in memory, which come with hyperparameters such as the learning rate, regularization strength, or the number of exemplars. However, these hyperparameters are usually tuned at the start and then kept fixed throughout the learning sessions, ignoring the fact that newly encountered tasks may have varying levels of difficulty. This study investigates the necessity of hyperparameter 'adaptivity' in Class-Incremental Learning: the ability to dynamically adjust hyperparameters such as the learning rate, regularization strength, and memory size according to the properties of the new task at hand. We propose AdaCL, a Bayesian Optimization-based approach to automatically and efficiently determine the optimal values for those parameters with each learning task. We evaluate the effectiveness of adaptivity on four different continual learning approaches and multiple datasets.

## 1   Introduction

This paper focuses on Class-Incremental Learning of deep neural network representations (Masana et al., 2020; De Lange et al., 2021). Unlike standard batch learning, which requires access to data from all categories simultaneously, Class-Incremental Learning can update a pre-trained deep classifier with new categories by expanding the classifier layer with new output nodes for new classes. This leads to more efficient learning and avoids the need to store task identities which allows for a more realistic scenario.

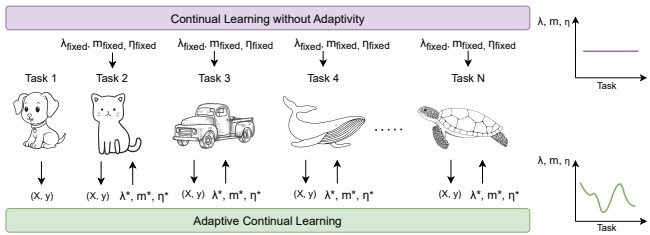

Figure 1: A comparison of fixed *vs.* adaptive continual learning (AdaCL). In this work, we explore the potential of tuning learning rate ($\eta$), regularization strength ($\lambda$) and memory size per task ($m$), allowing to learn adaptively.

While Class-Incremental Learning enables expanding a classifier without requiring task identities, it often results in a significant cost known as *catastrophic forgetting*. This occurs when the deep learner sacrifices accuracy on previously seen classes to learn new ones. Three major approaches have been explored to address this issue: regularization, replay and architecture adaptation. Regularization prevents abrupt shifts in the neural network weights while learning new classes (Kirkpatrick et al., 2017; Li and Hoiem, 2017). Replay stores a few exemplars per class in memory and replays them during new learning increments (Lopez-Paz and Ranzato, 2017). Architecture-based approaches build network structures by either expanding the existing network (Rusu et al., 2016; Yan et al., 2021) or by partially isolating network parameters to retain past class information (Liu et al., 2021a; Kang et al., 2022; Dekhovich et al., 2023). Although these methods improve the performance, they always use a fixed learning rate, regularization magnitude, and pre-defined memory size throughout the learning process.

This paper addresses the issue of dynamically adjusting *how much* to regularize, and store in Class-Incremental Learning for each new task. We explore whether adaptation is necessary for optimal performance, treating the learning rate, regularization magnitude, and memory size as latent variables that should be adjusted based on the current state of the learner and the complexity of the task (see Figure 1). We use Bayesian Optimization to efficiently discover the best hyperparameters per task. Our experiments on CIFAR-100 and MiniImageNet are expected to demonstrate that adapting these parameters to the tasks results in

significant improvement and give us new insight into how to adapt various new tasks. In summary, this paper makes the following contributions:

I. In this paper, for the first time, we raise the important issue of adaptive hyperparameter selection in class-incremental learning.

II. We propose to predict the learning rate, regularization magnitude, and memory size conditioned on the state of the deep learner and the current learning task via Bayesian Optimization.

III. Through large-scale experiments on well-established benchmarks, we plan to show that learning adaptively yields significant performance improvements, in terms of increasing accuracy while reducing forgetting.

## 2 Related Work

**Class-Incremental Learning.** Class-Incremental Learning updates a deep classifier with sequentially arriving data, usually with mutually exclusive categories (Masana et al., 2020; De Lange et al., 2021; Wang et al., 2023; Zhou et al., 2023; Kilickaya et al., 2023). However, when novel data arrives, previous training data becomes unavailable, leading to catastrophic forgetting. To mitigate this, researchers have developed three main approaches: (i) regularization-based methods, which stabilize important parameters or distill previous knowledge into the model (Kirkpatrick et al., 2017; Zenke et al., 2017; Lee et al., 2017; Li and Hoiem, 2017; Chaudhry et al., 2018a; Zhou et al., 2021b; Zhu et al., 2021), (ii) replay-based methods, which usually benefit from regularization-based methods and store a subset of training data to rehearse during learning (Rebuffi et al., 2017; Chaudhry et al., 2018b; Wu et al., 2019; Aljundi et al., 2019; Ostapenko et al., 2019; Xiang et al., 2019; Zhao et al., 2020; Liu et al., 2021b; Petit et al., 2023) and (iii) architecture-based methods designs network architectures by extending the network (Rusu et al., 2016; Yan et al., 2021; Zhu et al., 2022) or freezing network parameters partially to preserve old class knowledge (Liu et al., 2021a; Kang et al., 2022; Dekhovich et al., 2023).

However, current studies assume a constant amount of regularization and memory size per task throughout learning sessions which is unnatural, since learning unfamiliar objects requires more plasticity than learning familiar ones. To address this issue, we propose an adaptive method in which the regularization magnitude and memory size are automatically tuned within each incremental learning step.

**Hyperparameter Optimization.** Hyperparameter Optimization (HPO) aims to optimize the hyperparameters of a given deep learning model, including the learning rate, layer size, or balance of different loss functions. In this paper, our focus is on balancing the contribution of a standard classification and the regularization loss as well as memory size per task if applicable. To tackle the HPO problem, complex techniques such as bi-level optimization (Franceschi et al., 2018) or gradient-based optimization (Baydin et al., 2018) have been proposed. Bi-level optimizers alternate between optimizing neural network weights and tuning the hyper-parameters, while gradient-based methods treat the entire network weights as a hyper-parameter to be updated.

Several recent studies (Chaudhry et al., 2019; De Lange et al., 2021; Liu et al., 2023) share our core motivation by investigating the impact of hyperparameter optimization in evolving tasks. De Lange et al. (2021) adopt a two-stage strategy: They initially fine-tune the current task to identify the optimal learning rate with a grid search for maximum plasticity and peak accuracy. Then, in the second stage, they introduce a new thresholding hyperparameter to naively balance the plasticity and stability trade-off: They start with a high regularization strength and decay it when the performance of the current task is below the defined threshold. However, their approach follows a very naive search since they basically apply two consecutive grid searches to decide the optimum value. Moreover, they focused on Task-Incremental setup and did not consider the memory size in their search space.

Chaudhry et al. (2019) tunes the hyperparameters for the first $T$ tasks with a grid search and then uses the best-found values in the remaining tasks. However, it assumes that the initial few tasks are representative enough for the rest of the tasks which may not be realistic in most of the cases. Again, they worked on the Task-Incremental scenario and did not consider the memory size in their search space.

Liu et al. (2023) uses reinforcement learning in a Class-Incremental scenario to adaptively find the best hyperparameter values while learning the tasks. They hold a validation set, similar to our study, to estimate rewards by finding the best set of hyperparameters. However, its search space is limited to learning rate, regularization strength, and the type of classifier.

In this work, we propose Bayesian Optimization (Snoek et al., 2012) with Tree Parzen Estimator due to its effectiveness over multiple hyperparameters. We evaluate the generality of our approach by dynamically tuning the learning rate, regularization strength, and memory size.

## 3 Method

**Overview.** Class-incremental learning involves updating a neural network with new classes as it comes in. Specifically, the learner receives a sequence of learning tasks $\mathcal{T}_{1:t} = (\mathcal{T}_1, \mathcal{T}_2, ..., \mathcal{T}_t)$, each with a corresponding dataset $\mathcal{D}_{\mathcal{T}} = (x_{i,t}, y_{i,t})^{n_t}$ consisting of $n_t$ instances per task.

Each input pair $x_{i,t}, y_{i,t} \in \mathcal{X}_t \times \mathcal{Y}_t$ is sampled from an unknown distribution where $x_{i,t}$ is the sample and $y_{i,t}$ is the corresponding label. It's important to note that the learning tasks are mutually exclusive, i.e., $\mathcal{Y}_{t-1} \cap \mathcal{Y}_t = \emptyset$. When a new learning task arrives, a deep convolutional network is optimized to embed the input instance into the classifier space $f_\Theta : \mathcal{X}_t \to \mathcal{Y}_t$, where $\Theta$ represents the parameters of the learner.

The incremental learner has two goals: to effectively learn the current task (*plasticity*) while retaining performance on all previous tasks (*stability*). This can be accomplished by optimizing the following function where $CE(\cdot)$ represents the Cross-Entropy used in classification, and $Reg(\cdot)$ is a regularization term that penalizes abrupt changes in the neural network weights (Li and Hoiem, 2017; Kirkpatrick et al., 2017; Rebuffi et al., 2017; Zhao et al., 2020):

$$\mathcal{L} = CE(f(x_{i,t}), y_{i,t}) + \lambda \cdot Reg(\Theta) \qquad (1)$$

### 3.1 Foundational Models

To regularize the weights of the backbone and store a few exemplars per task, we experimented with four popular, well-established techniques: EWC (Kirkpatrick et al., 2017), LwF (Li and Hoiem, 2017) iCaRL (Rebuffi et al., 2017) and WA (Zhao et al., 2020). We tried to select baselines that complement each other and serve as strong baselines within the field of incremental learning (Table 1).

Table 1: Selected models to evaluate the impact of adaptivity in Class-Incremental Learning.

| method | prior-based | distillation-based | exemplar collection | classifier correction |
|---|---|---|---|---|
| EWC | ✓ | | | |
| LwF | | ✓ | | |
| iCaRL | | ✓ | ✓ | |
| WA | | ✓ | ✓ | ✓ |

**EWC.** Elastic Weight Consolidation (Kirkpatrick et al., 2017) is a weighted regularization approach. The authors argue that not all weights contribute equally to learning a new task and estimate the importance of each weight in minimizing the classification loss for the current task: $Reg(\Theta) = ||\mathcal{F}(\Theta - \Theta')||$, where $\Theta'$ is the model weights from the previous learning step, $\mathcal{F}$ is the Fisher matrix of the same size as the weight matrices $\Theta$, re-weighting the contributions of each weight to stabilize the important neurons per task.

**LwF.** Learning-without-Forgetting (Li and Hoiem, 2017) is a knowledge-distillation approach where the teacher branch is the model from the previous task, and the student branch is the current model. The aim is to match the activations of the teacher and student branches, either at the feature or logit layer. We found that logit-based distillation yielded better performance. Formally, LwF minimizes the

following objective where $f'$ is the model from the previous learning step, and $KL(p_1, p_2)$ is the KL-divergence between two probability distributions $p_1$ and $p_2$:

$$Reg(\Theta) = KL(f(x_{i,t}), f'(x_{i,t})) \qquad (2)$$

**iCaRL.** The Incremental Classifier and Representation Learning (Rebuffi et al., 2017) leverages a hybrid approach that involves two main components: exemplar-based memory which is carefully selected to maintain representation and a regularization. The exemplar-based memory module retains a subset of exemplar samples from previous tasks, representing important instances that encapsulate the learned knowledge. By utilizing exemplars, iCaRL ensures the model's ability to recognize and classify past instances while discriminating between learned and new classes. The distillation loss as in Eq. 2 used for regularization, enables knowledge distillation from previous models to guide the learning process for new tasks. This distillation process allows the model to align logits of new classes with already learned classes to mitigate catastrophic forgetting.

**WA.** The Maintaining Discrimination and Fairness in Class Incremental Learning (Zhao et al., 2020) is a method that consists of two phases: maintaining discrimination and maintaining fairness. The first phase is similar to the previously established method (Rebuffi et al., 2017). Their study demonstrates that knowledge distillation is not sufficient by itself to prevent the model to treat old classes and new classes fairly since there is a high tendency towards new classes in the classifier layer to minimize the Eq 2. Therefore, the second stage named Weight Aligning (WA) focuses on maintaining fairness to correct this classifier bias towards new classes. WA showed that it treats all classes fairly, and significantly improves the overall performance.

### 3.2 Constancy Assumption in Class Incremental Learning

The scalar parameter $\lambda$ balances the contribution of the classification and regularization loss functions. A large value of $\lambda$ ensures minimal weight updates, which can sacrifice learning on the current task. Conversely, a small $\lambda$ yields good performance on the current task but may sacrifice performance on previous tasks, exacerbating catastrophic forgetting. Similarly, requirement for a fixed or predetermined memory size per task may not always be necessary, as it depends on the new task and its relationship to previous tasks. Specifically,where the new task is highly similar to previous tasks, it is possible to retain past knowledge by storing only a small number of representative samples. Conversely, when the new task is significantly distinct, it is reasonable to store a larger number of examples in memory to prevent catastrophic forgetting while learning the new task. However, as a common practice, impor-

tant hyperparameters such as learning rate ($\eta$), regularization strength ($\lambda$), and memory size ($m$) are set to a fixed or pre-defined scalar value throughout all incremental learning sessions with $t \in \mathcal{T}_{1:t}$; such that $\eta_t = \eta_{t-1}$, $\lambda_t = \lambda_{t-1}$ or $\lambda_t = \frac{t*c}{(t*c)+c}$ where c is the number of classes per task. Similarly, $m_t = m_{t-1}$ or $m_t = \frac{M}{t}$ where M is the pre-defined total memory size.

We hypothesize that the assumption of *constant or pre-defined learning rate, regularization strength, and exemplar size per task* is unrealistic for building accurate life-long learning machines. Our reasoning is two-fold:

**Low Plasticity and High Stability.** The incremental learner may encounter a novel object that is highly familiar with the previously learned tasks. For example, it may encounter the category *dog* after observing many other animal categories, such as *cat, cow, bird*. In this case, the learner does not need to store many exemplars from previous tasks or to be too plastic, as it can quickly transfer knowledge from the previous tasks where it is similar to the human learning process and referred to *low road transfer* (Perkins and Salomon, 1992). Hence, no drastic updates to the learned filters are necessary.

**High Plasticity and Low Stability.** The incremental learner may encounter a novel object that is highly unfamiliar with the previous tasks. For example, it may encounter the category *car* after observing many other animal categories, such as *cat, cow, bird*. In this case, the learner would require more exemplars from previous tasks to preserve old knowledge and high plasticity to learn about the novel object with never-before-seen parts, such as wheels.

### 3.3 AdaCL: Adaptive Continual Learning

AdaCL defines the regularization magnitude and memory size per task as functions that consist set of incremental tasks, conditioned on the current learning task and all previous tasks. Formally, we define $\eta(t) = \eta_1, \eta_2, \ldots, \eta_{t-1}, \eta_t$, and $\lambda(t) = \lambda_1, \lambda_2, \ldots, \lambda_{t-1}, \lambda_t$, and $m(t) = m_1, m_2, \ldots, m_{t-1}, m_t$ where $\eta_t$, $\lambda_t$ and $m_t$ are predicted by minimizing the following optimization problem:

$$\arg\min_{\eta,\lambda,m} \mathcal{L}(\Theta; V_t) = \arg\min_{\eta,\lambda,m} \sum_{i=1}^{|V_t|} [CE(f(x_{i,t}; \Theta), y_{i,t}) \quad (3)$$

Here, $V_t$ is a randomly selected class-balanced subset of the current task and previous tasks that guide the model's adaptation with careful consideration of both new and previous tasks' characteristics and prevents bias over certain classes. $\mathcal{L}(\Theta; V_t)$ is the loss function where the learning rate $\eta$, the regularization coefficient $\lambda$, and memory size per task $m$ is determined by solving the optimization problem. Our adaptive approach, AdaCL (Algorithm 1), starts after the first task since it is just a standard batch learning. In the following tasks, it retains the model $\theta_{t-1}$ trained on

---

**Algorithm 1** AdaCL: Adaptive Continual Learning

**Require:**
  $\theta_{t-1}$          ▷ model from previous task
  $X_t$          ▷ dataset from new task
  $M_{t-1} = m_1, \ldots, m_{t-2}, m_{t-1}$    ▷ memory from old tasks
  $V_{t-1} = v_1, \ldots, v_{t-2}, v_{t-1}$    ▷ val. set from tasks seen so far
  $\eta_{space}$       ▷ search space for learning rate
  $\lambda_{space}$       ▷ search space for regularization
  $m_{space}$       ▷ search space for memory
  $configs, epochs$    ▷ # of configurations and epochs
1: $V_t = V_{t-1} \cup v_t \leftarrow X_t$
2: **for** $c = 1, \ldots, configs$ **do**
3:     $\eta_t \leftarrow \eta_{space}$        ▷ $\eta$ for new task
4:     $\lambda_t \leftarrow \lambda_{space}$        ▷ $\lambda$ for new task
5:     $M_t : m_t \leftarrow m_{space}$    ▷ memory with a size of $m_t$
6:     $D = X_t \cup M_{t-1} \cup M_t$   ▷ concat new data and memory
7:     **for** $e = 1, \ldots, epochs$ **do**
8:        Train Eq. 1 with $\theta_{t-1}$ and $D$
9:        Evaluate Eq. 3 with $V_t$
10:    **end for**
11: **end for**
12: **return** $\theta_t, V_t, \eta_t^*, \lambda_t^*, M_t^*$    ▷ new model with optimal learning rate, regularization strength and memory size

---

the previous task, receives current task data $X_t$, and creates a validation set $V_t$. Then, training data $D$ is constructed and trained with Eq. 1 after the configuration for $\eta_t$, $\lambda_t$ and $m_t$ is selected by Bayesian Optimization (see section 3.4). After each epoch, the selected configuration is evaluated on the validation set $V_t$ with Eq. 3. Subsequently, this process is repeated until reaching the total number of configurations. The optimal learning rate $\eta_t^*$, lambda $\lambda_t^*$, and memory size per task $m_t^*$ are determined based on the validation performance.

This approach allows us to automatically adjust the learning rate, regularization strength, and memory size per task according to the specific learning task based on the given loss function which lets the model find the degree of difficulty itself, avoiding the unrealistic assumption of a fixed learning rate, regularization strength, and memory size throughout the learning process.

### 3.4 Bayesian Optimization via Parzen Estimator

We optimize the objective function using multivariate tree-structured parzen estimators (TPE) (Bergstra et al., 2011). TPE builds a conditional probability tree that maps hyperparameters to their respective model performances. Then it can be used to guide a search algorithm to find the optimal set of hyperparameters for the given model. In this study, TPE is utilized as a search algorithm where it searches within the provided range for learning rate, regularization strength, and memory size per task and then searches for the best value by evaluating across accumulated validation set which consists of previous and new tasks throughout incremental learning sessions. Specifically, we are planning to build upon the implementation from the Optuna framework (Akiba et al., 2019).

## 4   Experimental Protocol

**Datasets.** In this paper, we experiment with **CI-FAR100** (Krizhevsky et al., 2009) and **MiniImageNet** (Vinyals et al., 2016). Each dataset exhibits objects from 100 different categories. We train all the models with 10 tasks, with 10 classes within each learning task on both CIFAR100 and MiniImageNet. Both datasets have 5000 training, and 1000 testing color images per learning task, each with $32 \times 32$ and $64 \times 64$ resolution for CIFAR100 and MiniImageNet respectively.

**Metrics.** We resort to the standard metrics for evaluation, accuracy (ACC) which measures the final accuracy averaged over all tasks, and backward transfer (BWT) which measures the average accuracy change of each task after learning new tasks:

$$ACC = \frac{1}{T} \sum_{i=1}^{T} A_{T,i} \qquad (4)$$

$$BWT = \frac{1}{T-1} \sum_{i=1}^{T-1} (A_{T,i} - A_{i,i}) \qquad (5)$$

where $A_{T,i}$ represents the testing accuracy of task $T$ after learning task $i$.

**Baselines.** EWC, LwF, iCARL, and WA are our direct baselines since we built our experiments on them. We also compare our common baseline results with OMDP (Liu et al., 2023). Finally, we select one recent memory-free work FeTrIL (Petit et al., 2023), and one recent memory-based work PODNet (Douillard et al., 2020) to provide comprehensive insights.

**Implementation Details.** We employ adaptive hyperparameter optimization on the methods discussed in section 3.2, and subsequently, we compare the adaptive variants of these methods with their fixed (original) versions. For the fixed versions, we provide default $\eta$, $\lambda$ and $m$ as it is used in PYCIL (Zhou et al., 2021a) framework.

We implement all the methods in PyTorch (Paszke et al., 2019). We use ResNet-32 as the backbone (He et al., 2016). We set the number of epochs to $E$ for each configuration but use the Successive Halving (Li et al., 2018) scheduler for a more efficient search. We use SGD optimizer with momentum parameter set to 0.9 and weight decay set to $5e^4$ for the first task and $2e^4$ for the rest of the tasks. The batch size is set to $B$. We run experiments on three different seeds and report their average. We store a small subset of the validation data from each incremental learning step in the memory to evaluate the search algorithm. The search space for learning rate and memory per task is set between [lower, upper] and $[0, m]$ respectively. Search space for $\lambda$ is determined based on the ablation experiments.

## 5   Conclusion

This study introduces the idea of adaptive learning rate, regularization and memory size in addressing the challenges of Class-Incremental Learning. These parameters are treated as tunable variables that can be adjusted according to the learner's current condition and the complexity of the task. Leveraging the power of Bayesian Optimization, the paper presents a methodology to predict the optimal values for these parameters in each learning task. By conducting experiments on well-established benchmarks, the study aims to showcase the remarkable enhancements in performance achieved through adaptive learning, resulting in improved accuracy and diminished forgetting. To improve this paper, as a future studies, cost of tuning hyperparameters can be minimized and also getting rid of validation set or finding better way to build it can be considerable.To further enhance the refinement of this paper, potential avenues for future investigation could involve the reduction of hyperparameter tuning costs and the exploration of alternative methods for constructing or optimizing the validation set.

To sum up, our study leads the way in introducing the concept of adaptive hyperparameter optimization in the realm of Class-Incremental Learning, with a mindful consideration of the limitations we've recognized. As the field advances, we anticipate that these insights will shape the evolution of advanced continual learning approaches, empowering deep neural networks to adapt to evolving datasets.

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
