# OpenReview forum: "AdaCL: Adaptive Continual Learning"
_continualai.org/CLAI/2023/Unconference_Preregistration_Track — 1st CLAI Unconf_

### Official Review · Reviewer_owKS · 2023-08-06
**Address the 'catastrophic forgetting' in CIL by adjusting the regularization strength and memory size.**

**Clarity:** 3
**Originality:** 3
**Soundness:** 3
**Significance:** 2
**Rating:** 6
**Confidence:** 4

**Review:**

In this paper, the authors proposed dynamically adjusting the regularization strength and memory size per task when learning a new task for class incremental learning.

**Strengths:**

Overall, the writing of this paper is clear, the motivation sounds good, and the method sounds solid.


**Weaknesses:**

- The paper lacks updated related work references and baseline comparisons, such as [1] 'RMM: Reinforced Memory Management for Class-Incremental Learning,' presented at NeurIPS 2021, and [2] 'Adaptive Aggregation Networks for Class Incremental Learning,' presented at CVPR 2021.

- Moreover, the cited works (EWC, LwF, iCaRL) do not reflect the current trends in the field.

- Additionally, the paper overlooks an important aspect of Class Incremental Learning literature, namely the set of memory-free approaches. While the paper primarily focuses on memory-based approaches, it is essential to acknowledge and discuss memory-free approaches as well.

**Questions:**

n/a

**Protocol:**

The experiments Protocol lacks insights and side-to-side comparisons with different Class-Incremental Learning (CIL) approaches.
- As previously mentioned, I recommend adding current trending CIL papers to the references, as the current list only includes papers before 2020.
- To provide more detailed insights, I recommend  the authors separating the experiment section into 3 different parts:1) compare with memory-based papers 2) compare regularization baseline papers. 3) Discuss and compare with zero-shot CIL papers and memory-free papers.
- I also recommend adding some interpretable visualizations. For example, constructing a small synthetic experimental with image dataset MNIST/CIFAR-10 for both the baselines and the proposed method.

---

### Official Review · Reviewer_6BER · 2023-08-19

**Clarity:** 3
**Originality:** 2
**Soundness:** 2
**Significance:** 3
**Rating:** 6
**Confidence:** 5

**Review:**

The goal of the paper is the study of bayesian optimization methods for continual learning. The hypothesis is that finding hyperparameters that evolve over time is better than fixing static parameters, which is the common practice.

Overall, my score is positive because I believe the results could be interesting for the community. However, I believe the experimental protocol needs to be extended as I ask below. I expect these additional experiments to be added in the final version of the paper.

**Strengths:**

The hypothesis is sound. At the same time, it is a bit underexplored in CL. The proposed method is a straightforward application of bayesian optimization.

**Weaknesses:**

- related work: The paper ignores related work such as [1], [2], [3]. [1] and [2] find adaptive parameters over time

method:
- the method is not really continual since it assumes the existence of the validation set over the entire stream. How do we create this set? why should we use this set for validation instead of doing more replay?
	- the paper should highlight the limitations of the method
- Alg.1 should add the memory buffer and the validation set to the input and outputs of the methods since they are both updated during CL.

[1] M. De Lange et al. “A Continual Learning Survey: Defying Forgetting in Classification Tasks.” TPAMI 2022
[2] Y. Liu et al. “Online Hyperparameter Optimization for Class-Incremental Learning.” AAAI ’23
[3] A. Chaudhry et al. “Efficient Lifelong Learning with A-GEM.”

**Questions:**

See weaknesses and protocol.

**Protocol:**

- The paper should test replay with class-balanced reservoir sampling. As of now, WA is the only replay-based method.
- it is a bit strange that one of the most important hyperparameters, the learning rate, is ignored by the HPO. How do the authors plan to set it? I would argue that their hypothesis is valid also for the learning rate.
- the protocol should compare against [1], [2], [3]
	- care should also be taken to give the same amount of compute to each method.
- (sec: implementation details) it seems the authors plan to use fix $\lambda$ and $m$ from previous papers. These values should be found again via the baselines HPO methods instead.

---

### Official Review · Reviewer_aaVd · 2023-08-21
**An interesting direction to explore for continual learning**

**Clarity:** 3
**Originality:** 3
**Soundness:** 3
**Significance:** 3
**Rating:** 7
**Confidence:** 4

**Review:**

In existing class-Incremental Learning works shch as weight regularization and exemplar-based methods,  the strength and exemplar counts  are fixed, which may not suit varying task complexities during incremental learning. The study introduces AdaCL, which uses Bayesian Optimization to dynamically adjust regularization strength  and memory bank.

**Strengths:**

This paper is well-writing and the proposed idea of adaptive continual learning is interesting and novel.

**Weaknesses:**

Recently, several non-exemplar class incremental learning methods (e.g. FrTrIL, PASS, IL2A, and SSRE in PYCIL codebase) have been proposed, and I suggest authors introduce them in the related work section.

**Questions:**

No question.

**Protocol:**

The experimental protocol is well designed. I additionally suggest authors include the experiments on ImageNet-1k.

---

### Decision · Program_Chairs · 2023-09-12

**Decision:**

Accept

**Comment:**

Dear authors,

Congratulations, your paper has been accepted at the ContinualAI Unconference 2023! We look forward to engaging in further discussions with you and others in the community.

Details will follow shortly regarding camera-ready versions. Please do take the feedback from reviews into account.

Thanks!